# Effect of Radiotherapy on Functional and Health-Related Quality of Life Outcomes after Jaw Reconstruction

**DOI:** 10.3390/cancers14194557

**Published:** 2022-09-20

**Authors:** Rebecca L. Venchiarutti, Masako Dunn, Emma Charters, Kai Cheng, Catriona Froggatt, Payal Mukherjee, Christine Wallace, Dale Howes, David Leinkram, Jasvir Singh, Kevin Nguyen, Tsu-Hui (Hubert) Low, Sydney Ch’ng, James Wykes, Raymond Wu, Jonathan R. Clark

**Affiliations:** 1Department of Head and Neck Surgery, Chris O’Brien Lifehouse, 119-143 Missenden Road, Camperdown, NSW 2050, Australia; 2Sydney School of Public Health, Faculty of Medicine and Health, The University of Sydney, Camperdown, NSW 2006, Australia; 3Royal Prince Alfred Institute of Academic Surgery, Sydney Local Health District, 145 Missenden Road, Camperdown, NSW 2050, Australia; 4Department of Oral Restorative Sciences, Westmead Centre for Oral Health, Corner of Hawkesbury Road and Darcy Road, Westmead, NSW 2145, Australia; 5School of Dentistry, Faculty of Medicine and Health, The University of Sydney, 2 Chalmers St., Surry Hills, NSW 2010, Australia; 6Sydney Medical School, Faculty of Medicine and Health, The University of Sydney, Camperdown, NSW 2006, Australia; 7Department of Otolaryngology—Head & Neck Surgery, Faculty of Medicine and Health Sciences, Macquarie University, Macquarie Park, NSW 2109, Australia; 8Department of Plastic Surgery, Royal Prince Alfred Hospital, 50 Missenden Road, Camperdown, NSW 2050, Australia; 9Department of Radiation Oncology, Chris O’Brien Lifehouse, 119-143 Missenden Road, Camperdown, NSW 2050, Australia

**Keywords:** head and neck cancer, jaw reconstruction, osteoradionecrosis, quality of life, speech, swallowing

## Abstract

**Simple Summary:**

Reconstructive surgery is critical to restore form and function after treatment for head and neck cancer (HNC). The aim of this cross-sectional study was to describe long-term quality of life (QoL) and functional outcomes among patients with a history of HNC who underwent reconstruction of the mandible and/or maxilla. Patients who had radiotherapy either before or after their index reconstruction reported significantly worse functional and QoL outcomes, including speech, swallowing, eating and drinking, appearance, smiling, and satisfaction with information. Swallowing, salivation, oral competence, and satisfaction with information worsened with increasing time since surgery. Women and younger patients also reported worse functional and QoL outcomes, especially speech and facial aesthetics. Understanding long-term outcomes of jaw reconstruction is important for both patients and clinicians to make evidence-based decisions about treatment options. We have identified several groups at risk of poorer outcomes that may benefit from enhanced pre-operative counselling and post-operative monitoring.

**Abstract:**

Long-term health-related quality of life (HRQOL) and functional outcomes following mandibular and maxillary reconstruction are lacking. To determine these outcomes, a cross-sectional study of patients with a history of cancer who underwent jaw reconstruction was undertaken. Participants were identified from a database of jaw reconstruction procedures at the Chris O’Brien Lifehouse (Sydney, Australia). Eligible patients had at least one month follow-up, were aged ≥18 years at surgery, and had history of malignancy. HRQOL was measured using the FACE-Q Head and Neck Cancer Module (FACE-Q H&N). Functional outcomes were measured using the FACE-Q H&N, MD Anderson Dysphagia Inventory (MDADI) and Speech Handicap Index (SHI). Ninety-seven questionnaires were completed (62% response rate). Mean age of respondents was 63.7 years, 61% were male, and 64% underwent radiotherapy. Treatment with radiotherapy was associated with worse outcomes across 10/14 FACE-Q H&N scales, three MDADI subscales and one composite score, and the SHI. Mean differences in scores between irradiated and non-irradiated patients exceeded clinically meaningful differences for the MDADI and SHI. Issues with oral competence, saliva, speaking, and swallowing worsened with increasing time since surgery. Younger patients reported greater concerns with appearance, smiling, speaking, and cancer worry. Women reported greater concerns regarding appearance and associated distress. History of radiotherapy substantially impacts HRQOL and function after jaw reconstruction. Age at surgery and gender were also predictors of outcomes and associated distress. Pre-treatment counselling of patients requiring jaw reconstruction may lead to improved survivorship for patients with head and neck cancer.

## 1. Introduction

Head and neck cancer (HNC) encompasses a spectrum of malignancies arising in the upper aerodigestive tract and skin, with varying degrees of biological aggressiveness and survival outcomes [1]. Historically a cancer with poor survival, the prognosis of several HNC subtypes has improved in recent decades owing to advances in detection and treatment options, as well as shifting aetiologies [2,3]. With increasing numbers of HNC survivors, ptimization of health-related quality of life (HRQOL) has joined survival as a key outcome of concern following treatment.

Treatment for HNC often requires a multimodal approach, including combinations of surgery, radiotherapy and/or systemic therapy to achieve optimal survival outcomes [4]. Ablative surgery for HNC can be extensive, often involving bone and/or soft tissue resection that results in defects requiring reconstructive surgery [5,6]. Reconstruction may also be required in cases of osteoradionecrosis of the jaw (ORN-J), of which incidence varies from 2 to 18.1% of people treated for HNC [7,8,9,10,11,12,13,14,15,16] and for which bone resection and dentoalveolar surgery are known risk factors [14,17]. The goal of reconstructive surgery is to Return form and improve function, which is particularly important for defects of the oral cavity where essential functions of speaking, swallowing, non-verbal communication, and breathing may be compromised [18].

Long-term data on functional and patient-reported HRQOL outcomes following reconstructive surgery of the jaw are lacking, which creates challenges in addressing supportive care needs in an evidence-based manner [19]. To date, most studies evaluating functional and/or HRQOL outcomes after jaw reconstruction surgery have been small, with limited follow-up [20,21,22,23,24,25]. Jacobsen and colleagues suggested that the persistent effects of (chemo)radiotherapy contribute to poor function and quality of life, despite the restoration of mandibular continuity after jaw reconstruction for osteoradionecrosis [22]. Speech and swallowing outcomes have been described as inferior among irradiated patients who underwent flap reconstruction after glossectomy compared to those who underwent flap reconstruction without irradiation [23,25,26].

The objective of the current study was to describe HRQOL and functional outcomes of individuals after reconstructive surgery of the maxilla and/or mandible at our institution. Our secondary objective was to identify treatment, sociodemographic, and clinical factors that predict HRQOL and functional outcomes. We hypothesised that patients with any history of head and neck irradiation (either before or after the index jaw reconstruction) would have inferior functional and HRQOL outcomes compared to patients who underwent jaw reconstruction and did not undergo radiotherapy.

## 2. Materials and Methods

### 2.1. Study Design and Setting

This was a cross-sectional survey of adult (≥18 years of age) individuals who underwent reconstructive surgery of the mandible and/or maxilla for primary or recurrent cancer or cancer treatment-related complications (osteoradionecrosis) at the Chris O’Brien Lifehouse/Royal Prince Alfred Hospital Head and Neck Service in Sydney, Australia, from 2008 to 2020. The interval between reconstructive surgery and completion of the questionnaire ranged between 1.3 and 142 months (median 2.64 months). Individuals were identified from a cross-sectional, prospectively maintained database and were excluded if they opted out of completing the survey or were living overseas at the time of the survey administration. Strengthening the Reporting of Observational studies in Epidemiology (STROBE) guidelines were followed in the reporting in this manuscript [27].

### 2.2. Survey Content

HRQOL and functional outcomes were measured using the FACE-Q Head and Neck Cancer Module (FACE-Q H&N), a 102-item questionnaire measuring functional, psychosocial, and experiential outcomes over 14 scales (eating, oral competence, salivation, speaking, swallowing, smile, appearance, drooling distress, eating distress, appearance distress, smile distress, speaking distress, cancer worry, and satisfaction with information) [28]. The FACE-Q H&N has demonstrated strong internal consistency and test–retest reliability; Cronbach’s alpha (α) >0.9 for 11 of the 14 scales and interclass correlation coefficients ranged from 0.86 to 0.98. Participants also completed the MD Anderson Dysphagia Inventory (MDADI), a well-validated widely used tool within HNC research to measure the impact of dysphagia on quality of life. The MDADI consists of 20 items, which measure an individual’s global, physical, emotional, and functional perceptions of swallowing ability. Nineteen of the items are summarised into a composite score (weighted average of the physical, emotional, and functional subscale questions). Cronbach’s alpha (α) of the MDADI subscales ranges from 0.85 to 0.93, demonstrating high internal consistency, and test–retest reliability ranges from 0.69 to 0.88 [29]. The Speech Handicap Index (SHI) was used to measure change relating to speech-related quality of life [30]. The SHI is a 30-item patient-reported outcome measure which generates a total score. The English translation has internal consistency reliability values (Cronbach’s alpha) of 0.98, 0.95 and 0.98 for the total SHI, speech subscale, and psychosocial subscale, respectively. Test–retest reliability scores are also high (Spearman’s rank correlation coefficient 0.92, 0.88 and 0.89 for the total SHI, speech subscale, and psychosocial subscale, respectively). The survey was administered either by mail or online, according to participant preference.

### 2.3. Treatment and Clinical Data

Data on past treatments (surgery, radiotherapy, chemotherapy), surgical information (use of virtual surgical planning, bone free flaps), demographics (age at surgery, gender), and comorbidities were extracted from medical records. For patients who receive radiotherapy, timing of irradiation was classified as prior to or after the index reconstruction for subgroup analysis.

### 2.4. Statistical Analysis

Descriptive statistics were used to report respondent characteristics. Raw scores for each scale of the FACE-Q H&N were summed to provide a total scale score, which were then converted to a score on a scale of 0 (worst outcome) to 100 (best outcome) [28]. The exception is the cancer worry scale, where a higher score reflects a worse outcome. Responses for each question of the MDADI subscales were summed, and a mean score calculated. This mean score was multiplied by 20 to obtain a score ranging from 0 (extremely low function) to 100 (high functioning) [29]. A total SHI score was calculated by summing all items for a score ranging from 0 to 120, with higher scores indicating higher levels of speech-related problems. Clinically meaningful differences between groups have been previous defined as ≥10 points for the MDADI [31] and 12 points for the SHI [30]. Linear regression analysis was used to analyse associations between HRQOL and functional outcomes with key demographic and clinical variables (time since surgery, presence of bony free flap, history of radiotherapy, gender, and age at surgery). Multivariable regression models were constructed adjusting for these variables. A *post hoc* subgroup analysis was also planned to determine whether timing of radiotherapy (either before or after the index reconstruction) impacted outcomes. This subgroup analysis followed the principles described above and excluded patients with no known history of radiotherapy. Analyses were performed using SAS Version 9.4 (SAS Institute Inc., Cary, NC, USA). Statistical significance was taken at *p* < 0.05.

## 3. Results

### 3.1. Participants, Response Rates, and Clinical Information

A total of 256 patients (undergoing 278 procedures) were assessed for eligibility, of which 99 patients were excluded as they were deceased (N = 49), declined (N = 10), or did not have a history of cancer (N = 40). Of the remaining 157 eligible patients, 97 (62%) returned the questionnaire.

Most respondents were male (N = 59, 61%) and the mean age of respondents was 63.7 years (SD 13.4 years) (Table 1). Over two-thirds (69%) had a history of squamous cell carcinoma, and 11% (N = 11) underwent a reconstruction for osteoradionecrosis. Most reconstructions were of the mandible alone (61%) and 28 patients (29%) underwent virtual surgical planning (VSP). Almost two-thirds of patients had a history of radiotherapy (N = 62, 64%)—19 prior to the index reconstruction, and 43 after the index reconstruction. Raw scores, presented as mean (standard deviation, SD) are presented in Table 2, alongside possible scores for each instrument and/or scale.

### 3.2. FACE-Q H&N

A history of radiotherapy (either before or after the index reconstruction) was a predictor of worse outcomes across most scales of the FACE-Q H&N (Table 3). The greatest mean differences (MD) between groups (radiotherapy versus no radiotherapy) were in speaking (MD −25.1 [95%CI −38.8 to −11.5]), facial appearance (MD −24.0 [95%CI −37.0 to −10.9]) and eating and drinking distress (MD −23.0 [95%CI −37.7 to −8.4]) scales. Patients with a history of radiotherapy reported significantly lower scores across all functional scales. Within QOL scales, only distress associated with eating and drinking, and speaking were significantly higher compared to those with no radiotherapy history. Cancer worry did not differ significantly between groups (MD 3.4 [95%CI −7.7 to 14.4]), however patients with a history of radiotherapy reported a worse satisfaction with information provided (MD −11.6 [95%CI −20.7 to −2.5]).

Scores in three functional scales decreased with each year since surgery: oral competence (MD −2.7 per year [95%CI −5.2 to −0.3]), saliva (MD −3.4 [95%CI −5.5 to −1.3) and swallowing (MD −2.4 [95%CI −4.5 to −0.3]) (Appendix A). Satisfaction with information also decreased with time from surgery (MD −1.9 per year [95%CI −3.6 to −0.2]). Younger age at surgery was associated with worse functional outcomes in the scales of appearance, smile and speaking, greater distress associated with smile and speaking, and higher cancer worry scores. Compared to males, females reported higher levels of concern about appearance (MD −16.0 [95%CI −28.5 to −3.6]), appearance-related distress (MD −19.3 [95%CI −33.1 to −5.4]), and drooling-related distress (MD −17.0 [95%CI −31.0 to −3.0]).

### 3.3. MDADI

A history of radiotherapy was an independent predictor of lower scores across all three subscales of the MDADI, as well as the composite score (Table 3). Mean differences ranged from −18.9 [95%CI −26.8 to −6.9] for the functional score subscale to −20.8 [95%CI −29.8 to −11.9] for the physical score subscale. Each of the mean differences exceeded the 10-point difference published by Hutcheson and colleagues [31] indicating clinically meaningful between-group differences in swallowing function. Increasing time between reconstruction surgery and completion of the questionnaire was associated with worse scores on the physical subscale of the MDADI (MD −1.8 per year [95%CI −3.6 to −0.06]) (Appendix A). However, time from reconstruction surgery to completion of the questionnaire did not influence scores for in the other subscales or global MDADI score.

### 3.4. SHI

A history of radiotherapy was an independent predictor of worse outcomes on the SHI (Table 3). The mean difference between groups was −19.3 (95%CI 7.1 to 31.5), which exceeded the 12-point difference published by Rinkel and colleagues [30] to indicate a clinically meaningful between-group difference. Younger age at surgery was associated with worse scores on the SHI (MD −0.6 per year [95%CI −1.0 to −0.2]) (Appendix A).

### 3.5. Subgroup Analysis—Timing of Radiotherapy

Subgroup analysis was performed on 62 patients who underwent radiotherapy either before (N = 19) or after (N = 43) the index reconstruction. This showed that only scores from the appearance scale of the FACE-Q H&N differed on multivariable analysis (Table 4). Patients who underwent radiotherapy before the index reconstruction reported worse appearance scores (MD −16.5 [95%CI −32.6 to −0.5]) compared to those who underwent radiotherapy after reconstruction. In addition, patients who had a history of radiotherapy prior to reconstruction had worse scores on the MDADI functional subscale (MD −10.8 [95%CI −23.5, 2.0]). This did not reach statistical significance but represents a clinically meaningful between-group difference. No other between-group differences reached statistical significance nor exceeded clinically meaningful differences.

## 4. Discussion

Radiotherapy is an essential component of HNC treatment. Whilst the impacts of radiotherapy on function and HRQOL are well documented in the HNC literature, there is limited understanding of the effects among the subset of HNC patients who undergo jaw reconstruction, a unique group for which optimal form and function are inextricably linked. The findings from the present study demonstrate a significant association between a history of radiotherapy and functional and HRQOL outcomes among patients who have undergone mandibular or maxillary reconstruction for head and neck malignancy or ORN-J. A clinically meaningful difference in swallowing outcomes was observed between irradiated and non-irradiated patients as measured by the MDADI composite score. Distress associated with eating and drinking (measured using the FACE-Q H&N) was significantly higher among irradiated patients. Speaking function, measured using the FACE-Q H&N and the SHI was significantly worse among irradiated patients, and associated distress was also higher. These findings are consistent with smaller series, which have reported high rates of morbidity and significant negative effects on functional and HRQOL outcomes in patients receiving post-operative radiotherapy (PORT) after free flap reconstruction [21,23,25]. The sub-group analysis of patients who underwent radiotherapy showed that only scores on the appearance scale of the FACE-Q H&N differed based on whether patients had radiotherapy prior to or after the index reconstruction.

Radiotherapy with or without concurrent chemotherapy is often used in the treatment of patients with oral cancers with adverse clinicopathological features. The benefit of post-operative radiotherapy (PORT) in patients with high-risk adverse features, such as involved margins, perineural invasion (PNI), multiple involved nodes, or extracapsular spread (ECS) is well established. However, the efficacy of PORT on local and regional control in patients with low or intermediate risk factors is less certain. In such cases, it is often clinician and/or patient preference that drives decision making about whether to undergo PORT, balancing the potential morbidity, risk of late effects, and impact on HRQOL with improvements in disease control. In the present study, 64% of patients underwent a course of radiotherapy. This reduced slightly after excluding patients who underwent reconstruction for ORN-J (51/86 patients, 59%) reflecting the case-mix of patients requiring both ablative and reconstructive surgery, among which many had locally advanced tumours with high-risk adverse features present. Most of the patients in our cohort received radiotherapy after the index reconstruction (n = 43, 69%). Of the 18 patients who had radiotherapy prior to their index reconstruction, radiotherapy would have either been delivered in the post-operative setting (after a previous ablative surgery) or as definitive treatment for head and neck cancer. Our institutional practice has been to recommend PORT in patients with any high-risk pathological feature or in those with multiple intermediate risk features, recognising that intermediate risk features in isolation have limited impact on mortality [32,33,34,35]. Omitting PORT where possible after primary ablative surgery may also increase options for surgical salvage, [36] reserving PORT for the management of locoregional recurrence, while avoiding early and late effects of irradiation.

In addition to radiotherapy, other independent predictors of functional and HRQOL outcomes were identified on multivariable analysis (Appendix A). Gender-based differences were evident for self-reported aesthetic outcomes (measured in the appearance scale of the FACE-Q H&N), with women reporting significantly worse satisfaction with appearance after reconstruction and greater appearance-related distress. This finding supports previous qualitative research investigating the information needs of HNC patients prior to surgery, in which elderly male patients stated that “appearance was of little consequence”, prioritising the potential cure of cancer from surgery over aesthetic outcomes. [37]. Though no gender differences were observed in functional scales of drooling and smiling, women reported greater distress associated with these outcomes compared to men. The greater distress and dissatisfaction with aesthetic outcomes that women reported aligns with a recent study on patient-reported outcomes after microvascular reconstruction for HNC [38]. The authors, who also used the FACE-Q H&N modules, concluded that women with a history of anxiety or depression, recurrent disease, or prolonged post-operative feeding tube requirements were most at risk of appearance-related psychosocial impairment. Similar gender disparities have also been observed in aesthetic outcomes among patients undergoing facial skin cancer surgery [39]. Altered perception of body image after head and neck surgery has implications for one’s ability to reintegrate into society, including engaging in employment [40], ultimately impacting long-term quality of life [41]. Declines in important head and neck functions, such as eating and drinking, oral competence, salivation, smiling, speaking, and swallowing were all associated with a history of radiotherapy in this study, and several of these have been identified as barriers in returning to work after HNC treatment in a study by Buckwalter and colleagues [42].

The results presented in this study add to the literature that may guide patient and clinician decision-making in the context of jaw reconstruction and radiotherapy to optimise patient satisfaction, particularly around function and HRQOL outcomes. Patients who undergo HNC treatment report feeling ill-prepared for alterations in their appearance, functional difficulties, and long-term adjustments that need to be made after surgery [37]. There is evidence demonstrating that psychosocial needs of patients treated for HNC are not being met, with disfigurement being a significant contributor to negative body image, worry, and social withdrawal [43]. While satisfaction with information received by survivors has been reported as high as 82.5% in one study [44], many patients still report unmet information and support needs, particularly in regards to emotional well-being, psychosexual health, and financial aspects of survivorship [45]. Satisfaction with pre-treatment information is an established predictor of psychological outcomes, with HNC patients who are more dissatisfied with information reporting greater depression post-treatment and scoring lower on the Mental Component Summary of the Short Form-12 (SF-12v2) Health Survey up to 6–8 months after treatment [46]. While it may not be possible to avoid radiotherapy when seeking to achieve the best oncological outcome, patients should be counselled of the benefits and disadvantages of PORT, a discussion that can be further informed by the data from the current study. Commencing pre-operatively, routine collection of patient-reported outcomes (including HRQOL and functional outcomes) should be standard of care for all patients who are scheduled to undergo radical resection with free flap reconstruction. Prospective, repeated data collection can aid monitoring of changes in function and HRQOL, which may be amenable to early intervention from clinical teams to prevent further deterioration and negative impacts on patient’s lives.

### Strengths and Limitations

The primary strength of this study is that it assesses a large cohort of consecutive patients who underwent jaw reconstruction surgery at a single institution managed by a core multidisciplinary team that was largely unchanged over the study period, resulting in a relative consistency in management approaches. The response rate of 62% among eligible patients is also high, particularly given approximately one-fifth of patients (30/157) were more than five-years from surgery and may not have been in routine follow-up. In addition, we utilised three well-validated survey instruments to comprehensively assess functional and HRQOL outcomes. While this resulted in some overlap in the questions being asked in each tool and the scales reported, we found consistency across certain scales that provides evidence for the reliability of the findings. For example, radiotherapy and age at surgery were independent predictors for both the SHI and the speech function scale of the FACE-Q H&N, and radiotherapy was a predictor of negative outcomes across all MDADI scales and all functional eating or swallowing scales of the FACE-Q H&N.

The limitations of this study include the design, which was retrospective and cross-sectional, which allows identification of associations between variables and outcomes, but not necessarily causation. While our multivariable analysis adjusted for time since reconstruction, conclusions related to temporal trends should still be drawn with caution, as we were unable to include data on recurrence, new primary tumours, or re-treatment in the analysis. Complete clinical, pathological, and treatment-related data were not available for all participants, and therefore we were only able to adjust for a limited number of factors in the multivariable models. The confounding effect of unmeasured factors that may impact on functional and HRQOL outcomes such as performance status, smoking history, pre-treatment functional status, tumour stage, disease relapse, reconstruction site, and use of salvage or subsequent therapies were not able to be accounted for. Lastly, while the cohort was drawn from a single institution with high surgical volume, this limits the external validity of the findings as institutional practices for reconstruction and radiotherapy likely vary between centres.

## 5. Conclusions

Despite efforts of multidisciplinary teams, patients requiring reconstructive jaw surgery after HNC may still experience negative long-term outcomes impacting their ability to fully reintegrate into society after treatment. Selected groups, such as those who undergo radiotherapy, women, and younger patients, appear at greater risk of having negative functional and HRQOL outcomes. The findings of this study may help guide pre-treatment counselling to prepare patients for this reality, which in turn may improve psychosocial outcomes and quality of life during the survivorship period.

## Figures and Tables

**Table 1 cancers-14-04557-t001:** Participant characteristics.

Characteristic	N (%)
**Gender**	
Male	59 (60.8)
Female	38 (39.2)
**Mean age, years (SD)**	63.7 (13.4)
**Diagnosis**	
Squamous cell carcinoma	67 (69.1)
Osteoradionecrosis	11 (11.3)
Osteosarcoma	5 (5.2)
Adenoid cystic carcinoma	2 (2.1)
Mucoepidermoid carcinoma	3 (3.1)
Other malignant diagnosis	9 (9.3)
**Reconstruction site**	
Maxilla	36 (37.1)
Mandible	59 (60.8)
Maxilla and mandible	2 (2.1)
**Time since reconstruction, years**	
Median (minimum, maximum)	2.64 (0.22, 11.83)
Mean (SD)	3.32 (2.47)
**Virtual surgical planning**	
Yes	28 (28.9)
No	69 (71.1)
**Bone free flap**	
Yes	53 (54.6)
*Fibula free flap*	*44 (45.4)*
*Scapula free flap*	*6 (6.2)*
*Deep circumflex internal artery (DCIA) free flap*	*3 (3.1)*
No	44 (45.4)
*Radial forearm free flap*	*20 (20.6)*
*Anterolateral thigh free flap*	*20 (20.6)*
*Other free flap **	*4 (4.1)*
**Radiotherapy**	
Yes	62 (63.9)
*Prior to index reconstruction*	*19 (19.6)*
*After index reconstruction*	*43 (44.3)*
No	33 (34.0)
Unknown	2 (2.1)
**Chemotherapy**	
Yes	22 (22.7)
No	68 (70.1)
Unknown	7 (7.2)
**History of diabetes**	
Yes	7 (7.2)
No	77 (79.4)
Unknown	13 (13.4)
**History of vasculopathy**	
Yes	10 (10.3)
No	67 (69.1)
Unknown	20 (20.6)
**History of smoking**	
Yes	40 (41.2)
No	43 (44.3)
Unknown	14 (14.4)

SD = standard deviation. * Other free flaps included superficial circumflex iliac artery perforator (SCIP) flap, ulnar forearm free flap, and obturator.

**Table 2 cancers-14-04557-t002:** Raw scores from cross-sectional survey of patients using the FACE-Q H&N Module, MD Anderson Dysphagia Inventory, and Speech Handicap Index. Higher scores indicate better outcomes, except for the FACE-Q H&N Worry Scale and for the SHI Total Score, where higher scores indicate greater worry and greater levels of speech-related problems, respectively.

Instrument	N	Possible Score Range	Mean Score (SD)
**FACE–Q H&N Scale**			
*Appearance*	95	0–100	63.77 (31.95)
*Eating and drinking*	95	0–100	55.77 (22.44)
*Oral competence*	96	0–100	59.76 (30.29)
*Saliva*	96	0–100	67.00 (26.98)
*Smile*	95	0–100	65.59 (28.04)
*Speaking*	95	0–100	59.11 (32.98)
*Swallowing*	96	0–100	68.33 (26.44)
*Appearance distress*	95	0–100	64.37 (33.67)
*Drooling distress*	95	0–100	69.82 (33.64)
*Eating and drinking distress*	93	0–100	54.25 (33.39)
*Smiling distress*	93	0–100	74.33 (29.13)
*Speaking distress*	93	0–100	68.27 (26.60)
*Worry*	92	0–100	30.36 (24.84)
*Information*	87	0–100	78.69 (20.18)
**MDADI**			
*Global score*	92	0–100	69.35 (28.82)
*Emotional score*	92	20–100	72.87 (21.48)
*Functional score*	92	20–100	72.26 (22.52)
*Physical score*	92	20–100	70.33 (21.49)
*Composite score*	92	20–100	71.57 (20.82)
**SHI**			
*Total score*	91	0–120	28.76 (28.44)

Abbreviations: MDADI = MD Anderson Dysphagia Inventory; SHI = Speech Handicap Index; SD = standard deviation.

**Table 3 cancers-14-04557-t003:** Mean scores on the FACE-Q H&N, MDADI, and SHI based on whether patients did or did not have a history of radiotherapy. Higher scores indicate better outcomes, except for the FACE-Q H&N Worry Scale and for the SHI Total Score, where higher scores indicate greater worry and greater levels of speech-related problems, respectively.

Outcome	Radiotherapy Group	No Radiotherapy Group	Mean Difference *	Standard Error	95% CI	*p*-Value
** *FACE−Q H&N Scale* **						
*Facial appearance*						
Facial appearance	53.34	77.33	−24.00	6.56	−37.03, −10.94	<0.001
*Facial function*						
Eating and drinking	48.31	65.67	−17.36	4.81	−26.91, −7.80	<0.001
Oral competence	53.64	68.64	−15.01	6.52	−28.00, −2.05	0.024
Salivation	58.63	78.52	−19.89	5.62	−31.05, −8.73	<0.001
Smiling	57.15	75.73	−18.57	6.01	−30.52, −6.63	0.003
Speaking	50.00	75.12	−25.13	6.87	−38.78, −11.48	<0.001
Swallowing	60.28	80.59	−20.31	5.58	−31.39, −9.23	<0.001
*Quality of life scales*						
Appearance distress	58.76	69.81	−11.05	7.35	−25.65, 3.54	0.136
Drooling distress	63.22	76.69	−13.46	7.41	−28.19, 1.26	0.082
Eating and drinking distress	44.33	67.36	−23.03	7.38	−37.70, −8.37	0.002
Smiling distress	68.72	79.17	−10.45	6.37	−23.12, 2.21	0.105
Speaking distress	62.47	77.62	−15.15	5.81	−26.70, −3.59	0.011
Worry	32.02	28.63	3.39	5.56	−7.67, 14.45	0.544
*Experiential scale*						
Satisfaction with information	73.97	85.56	−11.60	4.56	−20.67, −2.52	0.013
** *MDADI* **						
*Global score*	59.47	85.15	−25.67	6.24	−38.10, −13.25	<0.001
*Emotional score*	64.96	84.83	−19.88	4.66	−29.14, −10.61	<0.001
*Functional score*	65.40	82.27	−18.87	5.01	−26.82, −6.91	0.001
*Physical score*	62.13	82.93	−20.81	4.50	−29.75, −11.86	<0.001
*Composite score*	63.02	81.88	−19.45	4.45	−28.30, −10.61	<0.001
** *SHI* **						
*SHI total score*	36.36	17.03	19.33	6.13	7.13, 31.53	0.002

* Adjusted for time since surgery, bone free flap, gender, and age at surgery. Abbreviations: MDADI = MD Anderson Dysphagia Inventory; SHI = Speech Handicap Index; CI = confidence interval.

**Table 4 cancers-14-04557-t004:** Subgroup analysis examining the effect of timing of radiotherapy on health-related quality of life and functional outcomes measured using the FACE-Q H&N, MDADI and SHI (N = 62). Higher scores indicate better outcomes, except for the FACE-Q H&N Worry Scale and for the SHI Total Score, where higher scores indicate greater worry and greater levels of speech-related problems, respectively.

Outcome	Radiotherapy Prior to Index Reconstruction	Radiotherapy after Index Reconstruction	Mean Difference *	Standard Error	95% CI	*p*-Value
**FACE−Q H&N**						
Facial appearance						
*Facial appearance*	42.80	59.33	−16.53	8.0	−32.55, −0.52	0.043
Facial function						
*Eating and drinking*	44.84	49.52	−4.68	5.0	−14.78, 5.42	0.36
*Oral competence*	48.91	55.25	−6.33	7.7	−21.72, 9.05	0.41
*Salivation*	56.41	60.03	−3.62	6.8	−17.27, 10.03	0.60
*Smiling*	53.65	58.71	−5.05	7.0	−19.05, 8.94	0.47
*Speaking*	40.26	54.79	−14.53	8.1	−30.69, 1.63	0.08
*Swallowing*	51.90	63.57	−11.67	6.3	−24.28, 0.94	0.07
Quality of life scales						
*Appearance distress*	51.67	62.48	−10.81	8.6	−28.08, 6.46	0.21
*Drooling distress*	56.98	65.62	−8.64	9.7	−28.02, 10.74	0.38
*Eating and drinking distress*	36.37	48.03	−11.66	9.0	−29.78, 6.46	0.20
*Smiling distress*	70.46	69.33	1.13	7.6	−14.03, 16.30	0.88
*Speaking distress*	54.61	67.56	−12.95	6.7	−26.33, 0.43	0.06
*Worry*	33.63	30.69	2.94	7.3	−11.75, 17.63	0.69
Experiential scale						
*Satisfaction with information*	71.26	76.56	−5.30	6.2	−17.75, 7.14	0.40
**MDADI**						
Global score						
Emotional score	58.82	68.27	−9.45	5.7	−20.81, 1.90	0.10
Functional score	58.02	68.78	−10.76	6.4	−23.53, 2.02	0.10
Physical score	55.98	64.91	−8.93	5.1	−19.20, 1.34	0.09
Composite score	57.34	66.92	−9.58	5.3	−20.17, 1.00	0.08
**SHI**						
SHI total score	54.51	63.19	−8.68	8.1	−24.84, 7.48	0.29

* Adjusted for time since surgery, bone free flap, gender, and age at surgery. Abbreviations: MDADI = MD Anderson Dysphagia Inventory; SHI = Speech Handicap Index; CI = confidence interval.

## Data Availability

The data presented in this study are available on request from the corresponding author. The data are not publicly available due to ethical and privacy restrictions.

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
