# Peer review of "Effect of Radiotherapy on Functional and Health-Related Quality of Life Outcomes after Jaw Reconstruction"

_cancers, 2022, doi:10.3390/cancers14194557_

Round 1
Reviewer 1 Report
97 - specify the median interval between reconstructive surgery and questionnaire completion, not only range.
100 - exclusion criteria: were patients with recurrent disease and ORN excluded? The same for 145 - were recurrent disease and ORN included in clinical variables? (you mentioned this question on 341 - but it should be clearly stated in the inclusion/exclusion criterias).
Please, consider as a part of discussion a recommendation to complete FACE-Q H&N/MDADI in all patients with ORN as a part of preparation for radical resection with fibula flap reconstruction and then to repeat these tests after the surgery - similar to Jacobson AS, Zevallos J, Smith M, Lazarus CL, Husaini H, Okay D, Buchbinder D, Persky M, Urken ML. Quality of life after management of advanced osteoradionecrosis of the mandible. Int J Oral Maxillofac Surg. 2013 Sep;42(9):1121-8. doi: 10.1016/j.ijom.2013.03.022. Epub 2013 May 1. PMID: 23643291.
Reference No 7 (Nabil S, Samman N. Risk factors for osteoradionecrosis after head and neck radiation: a systematic review. Oral Surg Oral Med 387 Oral Pathol Oral Radiol. 2012;113:54-69) is formally correct, but in real life the incidence of ORN is much higher, f/e:
Beadle et al = 16.1%
Beadle BM, Liao KP, Chambers MS, Elting LS, Buchholz TA, Kian Ang K, at al. Evaluating the impact of patient, tumour, and treatment characteristics on the development of jaw complications in patients treated for oral cancers: a SEER-Medicare analysis. Head Neck. 2013 Nov;35(11): 1599-605. Doi: 10.1002/hed.23205. Epub 2012 Nov 14
Sathasivam et al. = 18.1%
Sathasivam HP, Davies GR, Boyd NM. Predictive factors for osteoradionecrosis of the jaws: a retrospective study. Head Neck. 2018 Jan;40(1):46-54. Doi: 10.1002/hed.24907. Epub 2017 Nov 17), Tsai et al =7.5% (Tsai CJ, Hofstede TM, Sturgis EM, Garden AS, Lindberg ME, Wei Q, Tucker SL, Dong L. Osteoradionecrosis and radiation dose to the mandible in patients with oropharyngeal cancer. Int J Radiat Oncol Biol Phys. 2013 Feb 1;85(2):415-20. Doi: 1016/j.ijrobp.2012.05.032. Epub 2012 Jul 12)
Owosho et al = 4.3%
Owosho AA, Tsai CJ, Lee RS, Freymiller H, Kadempour A, Varthis S, at al. The prevalence and risk factors associated with osteoradionecrosis of the jaw in oral and oropharyngeal cancer patients treated ith intensity-modulated radiation therapy: The Memorial Sloan Kettering Cancer Center Experience.
Caparotti et al = 6.0%
Caparrotti F, Huang SH, Lu L, Bratman SV, Ringash J, Bayley A, at al. Osteoradionecrosis of the mandible in patients with oropharyngeal carcinoma treated with intensity-modulated radiotherapy. Cancer, 2017 Oct 1;123(19):3691-3700. Doi: 10.1002/cncr.30803. Epub 2017 Jun 13. Oral Oncol. 2017 January; 64:44-51. Doi:10.1016/j.oraloncology.2016.11.015.
Kojima et al = 7.7%
Kojima Y, Yanamoto S, Umeda M, Kawashita Y, Saito I, Hasegava T, at al. Relationship between dental status and development of osteoradionecrosis of the jaw: a multicentre retrospective study. Oral Surg Oral Med Oral Pathol Oral Eadiol 2017;124:139-145.
Moon et al = 5.5%
Moon DH, Moon SH, Wang K, Weissler MC, Hackman TG, Zanation AM, at al. Incidence of, and risk factors for, mandibular osteoradionecrosis in patients with oral cavity and oropharynx cancers. Oral Oncol. 2017 Sep; 72:98-103. Doi: 10.1016/j.oraloncology.2017.07.014. Epub 2017 Jul 16.
Aarup-Kristensen et al = 4.6%
Aarup-Kristensen S, Hansen CR, Fornber L, Brink C, Eriksen JG, Johansen J. Osteoradionecrosis of the mandible after radiotherapy for head and neck cancer: risk factors and dose-volume correlations. Acta Oncologica https://doi.org/10.1080/0284186X.2019.1643037
Also, you should mention that bone surgery is a recognised risk factor for development of ORN.
Reviewer 2 Report
The authors retrospectively investigated the HRQOL of a cohort of head and neck cancer patients after jaw reconstruction.
Radiotherapy was found as main factor influencing patients' HRQOL. HRQOL was measured using the FACE-Q, MDADI and SHI questionnaires.
Response rate was 62% which is really high.
In my opinion the authors present a well-designed and performed study. The manuscript has a clear structure and a concrete message.
Of couse the negative impact of radiotherapy is anything but new.
Interesting, in my view, is the result that radiotherapy before reconstruction shows worse outcome than radiotherapy after reconstruction. This supports the idea of primary reconstruction during ablative tumor surgery.
As a surgeon I miss a little bit more about the surgical aspects. The authors write: "No other between-group differences reached statistical significance nor exceeded clinically meaningful differences"
Nevertheless it would be interesting how for example virtual planning performed vs free hand bony reconstruction. Or maxillary reconstruction vs mandibular reconstruction. Also in my experience it is a big difference if you reconstruct a chin defect (far worse) or lateral corpus. Did the authors note the location of jaw reconstruction. Was there a difference between fibula, scapula or DCIA? Anyway I would appreciate more information about the used flaps. 54.6% were bony flaps. What types of flaps were the others?
Round 2
Reviewer 2 Report
It is a pity that the few recommendations were widely neglected although I am convinced that comparison of at least some of the mentioned parameters could be done within one day. And even when there is no significance of the results it would have been worth to do it for the first revision in a journal with an impact factor of 6.5.